



# Do Wind Turbines Pose Roll Hazards to Light Aircraft?

Jessica M. Tomaszewski[1], Julie K. Lundquist[1,2], Matthew J. Churchfield[2], and Patrick J. Moriarty[2]

[1]Department of Atmospheric and Oceanic Sciences, University of Colorado, Boulder, CO 80309-0311, USA
[2]National Wind Technology Center, National Renewable Energy Laboratory, Golden, CO 80401-3305, USA

**Correspondence:** Jessica Tomaszewski (jessica.tomaszewski@colorado.edu)

**Abstract.** Wind energy accounted for 5.6% of all electricity generation in the United States in 2016. Much of this development has occurred in rural locations, where open spaces favorable for harnessing wind also serve general aviation airports. As such, nearly 40% of all U.S. wind turbines exist within 10 km of a small airport. Wind turbines generate electricity by extracting momentum from the atmosphere, creating downwind wakes characterized by wind-speed deficits and increased turbulence.

Recently, the concern that turbine wakes pose hazards for small aircraft has been used to limit wind farm development. Herein, we assess roll hazards to small aircraft using large-eddy simulations of a utility-scale turbine wake. Wind-generated rolling moments on hypothetical aircraft transecting the wake in stably and neutrally stratified conditions are calculated. In both cases, only 0.001% of rolling moments experienced by hypothetical aircraft during down-wake and cross-wake transects lead to an increased risk of rolling.

*Copyright statement.* TEXT

## 1 Introduction

Due to its appeal as a renewable, low-carbon energy source, wind energy development has increased rapidly, accounting for 5.6% of electricity generation in the United States at the end of 2016 (EIA, 2017). Wind turbines generate electricity by extracting momentum from the atmosphere, thereby creating wakes characterized by a wind-speed deficit and increased

turbulence downwind (Lissaman, 1979; Sforza et al., 1981; Baker and Walker, 1984). As the size of turbine rotors increases, turbine wakes also increase, which may pose a greater hazard to nearby flying aircraft (Wang et al., 2015). Therefore, improved understanding of turbine wake characteristics is crucial to assess hazards to aircraft posed by wakes.

Numerous studies have investigated the structure, duration, and decay of wind turbine wakes. Most of these studies address wake impacts on surface temperature (Zhou et al., 2012; Rajewski et al., 2013; Smith et al., 2013) or surface fluxes (Rajewski

et al., 2014, 2016). Detailed measurements of winds in turbine wakes have been taken with lidars (e.g., Käsler et al. (2010); Iungo et al. (2012); Rhodes and Lundquist (2013); Aitken et al. (2014a)). Wind tunnel experiments have also been used to further aerodynamic research on wind turbines (Vermeer et al., 2003; Snel et al., 2007; Chamorro and Porté-Agel, 2009; Yang et al., 2012; Hancock and Zhang, 2015). Scaling such experiments to utility-scale turbines can be challenging: wind tunnel measurements typically cannot account for complexities in atmospheric profiles, such as inversions and wind veer.





Computational fluid dynamics (CFD) simulations of wind turbine wakes provide insights into turbine wakes (Sanderse et al., 2011). Both actuator-disk (Vermeer et al., 2003; Calaf et al., 2010; Mirocha et al., 2014; Vanderwende et al., 2016) and actuator-line (Sørensen and Shen, 2002; Troldborg et al., 2010; Churchfield et al., 2012; Marjanovic et al., 2017) methods have been used to simulate the rotor and downstream flow to study wake structure.

5   Extensive wind development occurs in rural locations, where open spaces favorable for wind energy are also home to numerous general aviation airports. Almost 40% of all wind turbines in the United States exist within 10 km of a small airport, and about 5% exist within 5 km of a small airport (Figure 1, based on data from OurAirports (2016); USGS (2014)). Currently, neither the Federal Aviation Administration (FAA) of the United States nor the Civil Aviation Authority (CAA) of the United Kingdom have detailed recommendations on the effects of wind-turbine-induced roll hazards for aircraft (CAA, 2012; Ho 10 et al., 2014).

The roll hazard metric most concerning to general aviation pilots is the rolling moment (Wang et al., 2015). The rolling moment is the aerodynamic force applied at a distance from an aircraft's center of mass that causes the aircraft to undergo angular acceleration about its roll axis. The roll axis is the longitudinal axis, running from the nose to the tail of the aircraft. Other wake hazards of concern to pilots are those generated from the wings of preceding aircraft during take-off as a consequence of 15 lift (Holzäpfel et al., 2007; Gerz et al., 2009).

Previous work has argued that turbine wakes present a serious roll hazard to general aviation aircraft. Mulinazzi and Zheng (2014) used a helical vortex model to represent a wind turbine wake from which aircraft roll hazards were calculated. The helical vortex model was scaled up from a wind tunnel study using miniature turbines with a rotor diameter of 0.254 m. The 4 m s$^{-1}$ inflow was meant to trigger transient helical tip vortices (Yang et al., 2012). Mulinazzi and Zheng (2014) scaled up 20 these wind tunnel results to the atmosphere by assuming comparable near- and far-wake turbulent flow structures between the tip vortices measured in the wind tunnel and the full wake produced in a real atmosphere by a utility-scale turbine. With calculations derived from this scaling, they suggest wind turbine wakes pose a significant roll hazard to general aviation aircraft at downwind distances as far as 4.57 km (2.84 miles) (Mulinazzi and Zheng, 2014). These findings have been used in multiple states to limit wind energy development. Reid Bell, airport manager at Pratt County Airport in Kansas, confirmed that the 25 Pratt wind farm project was relocated further away from the airport as a direct result of the Mulinazzi and Zheng (2014) study (Williams, 2014). The study has been used as a warning to aviators in Virginia as well (Hamilton, 2014).

Other researchers have also investigated the hazards that wind turbines pose to aircraft by computing the roll hazards from analytic representations of wakes. Wang et al. (2015, 2017) computed a wake wind field derived using the Beddoes circulation formula (Madsen and Rasmussen, 2004), proven to adequately match lidar field observations made of a wind turbine wake. 30 For a 30-m rotor diameter wind turbine, Wang et al. (2015, 2017) find the wake does not pose any roll hazards for aircraft 5 rotor diameters downstream from the turbine.

Rather than approximating wakes, we seek to explicitly resolve turbine wakes within a dynamic, non-stationary atmosphere. We therefore use large-eddy simulations (LES) to assess wind-generated roll hazards to small aircraft from the wake of a utility-scale wind turbine. Section 2 provides details on our simulations and methodology for analyzing the model data. Section



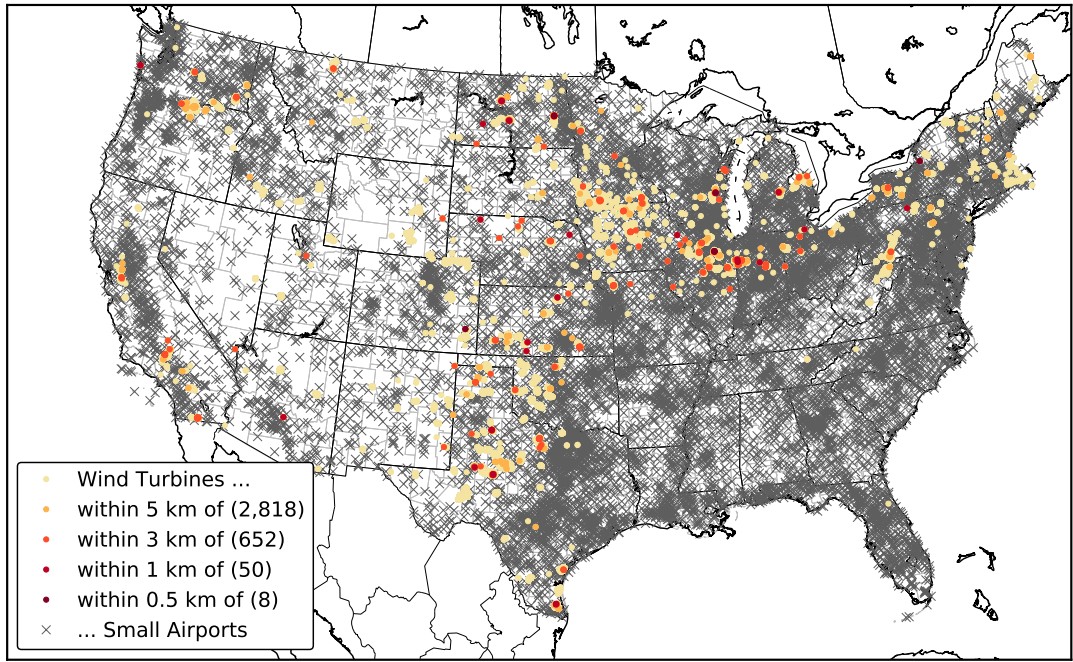

**Figure 1.** Map of wind turbines (dots) and small airports (x) in the continental United States. Turbines dots are colored increasingly red as their proximity to a small airport increases. The number of turbines in each distance range is shown in parentheses.

3 presents the results of our analysis, specifically the quantified roll hazards on hypothetical aircraft. We conclude in Section 4 with a discussion of our results and suggestions for future work.

## 2 Methods

### 2.1 Simulations

5  LES is a well-established method for studying wind turbine wakes. LES have been used to investigate wind turbine wake turbulence (Churchfield et al., 2012; Jha et al., 2015), the wind-farm boundary layer (Calaf et al., 2010), and the evolution of the wake structure in variable stability conditions and throughout the diurnal cycle (Lu and Porté-Agel, 2011; Mirocha et al., 2014; Aitken et al., 2014b; Bhaganagar and Debnath, 2014; Mirocha et al., 2015; Abkar et al., 2016; Englberger and Dörnbrack, 2018). Jimenez et al. (2007) find good agreement between LES and experimental data of turbine wakes, which has

10  led to the use of LES to correct for instrumentation error (e.g., Lundquist et al. (2015)).

Representing turbines in LES can be done via actuator disks, where the turbine rotor is represented by a permeable circular disk with uniformly distributed thrust forces (Vermeer et al., 2003; Calaf et al., 2010; Mirocha et al., 2014; Aitken et al., 2014b; Vanderwende et al., 2016), or by actuator lines, which represent the turbine blades as separate rotating lines (Sørensen and



Shen, 2002; Porté-Agel et al., 2011; Nilsson et al., 2015). Martínez-Tossas et al. (2015) compare actuator-line and actuator-disk models and conclude that they produce similar wake profiles; however, the actuator-line model can generate fine flow structures near the blades such as root and tip vortices that the actuator-disk model cannot.

We perform our simulations using the Simulator fOr Wind Farm Applications (SOWFA; Fleming et al. (2013); Churchfield and Lee (2014)). SOWFA is a computational fluid dynamics solver coupled with a turbine dynamics model. The LES solver is based on the Open Field Operations and Manipulation (OpenFOAM) toolbox version 2.4.x (OpenCFD, 2016). OpenFOAM is a set of C++ libraries and applications for use in solving the partial differential equations describing fluid flow. The simulations utilize the same two-step methodology as described in Churchfield et al. (2012), briefly summarized here. First, a precursor LES generates turbulent atmospheric flow on a domain with idealized periodic lateral boundaries. Once the turbulent boundary layer reaches quasi-equilibrium, a plane of turbulent data from the upwind lateral boundary is saved at each time step to be used as inflow boundary conditions for the simulation with the turbine. Next, a turbine is introduced into the flow initialized from the quasi-equilibrium precursor flow field. The side boundaries remain periodic, but the saved velocity and temperature data from the precursor are used as inflow Dirichlet boundary conditions for this simulation. The outflow boundary uses Neumann, zero-normal-gradient conditions. Like the precursor, the overall domain size is 3 km x 3 km in the horizontal and 1 km in height, but variable resolution is used. Most of the domain has 10-m resolution, like the precursor, but around the turbine and its wake, the resolution is gradually refined to 1.25 m (Figure 2). The time step in the wind turbine simulation is restricted to capture the motion of the blade tips.

The turbine model consists of an actuator-line representation of turbine blades (Sørensen and Shen, 2002). We model the GE 1.5-MW SLE wind turbine (Mendoza et al., 2015). This horizontal-axis, upwind turbine has a three-bladed rotor 77 m in diameter, with a hub height of 80 m. As one of the most widely deployed turbines worldwide, the wake effects on general aviation aircraft explored herein will be broadly applicable to numerous wind farms.

A subvolume of the flow surrounding the turbine and wake was sampled at 1 Hz. This sampled subvolume uses a Cartesian coordinate system, within which positive x (y) corresponds to downwind (cross-wind), and positive z corresponds to height above the surface. In all cases, the sampled subvolume extends 11 rotor diameters (D) in the x- (downwind) direction, 4D in the y- (cross-wind) direction, and 3D in the vertical direction. The computational grid resolution is approximately 1.25 m, uniform in all directions (Figure 2). This high resolution is necessary to adequately resolve flow structures that impact small (10-m) general aviation aircraft.

Previous observations note turbine wakes tend to diffuse more rapidly in convective conditions because of mechanical mixing by the surrounding air eroding the wake (Baker and Walker, 1984; Magnusson and Smedman, 1994; Bhaganagar and Debnath, 2014; Mirocha et al., 2015). We thus hypothesize that stable conditions present a worst-case scenario for general aviation aircraft due to longer-persisting wakes permitted by the reduced ambient atmospheric turbulence. As such, we simulate a neutral case and a stable case. Each simulation is spun up for 30,000 s, after which 100 s of data are output. The neutral case has a hub height inflow wind speed of 7.4 m s$^{-1}$ and a constant potential temperature profile of 300 K, while the stable case has a hub height inflow wind speed of 9.4 m s$^{-1}$ and potential temperature lapse rate of $-0.024$ K m$^{-1}$ over the vertical extent of the domain. These wind speeds were selected because of their location on the power curve: above the cut-in wind speed (3




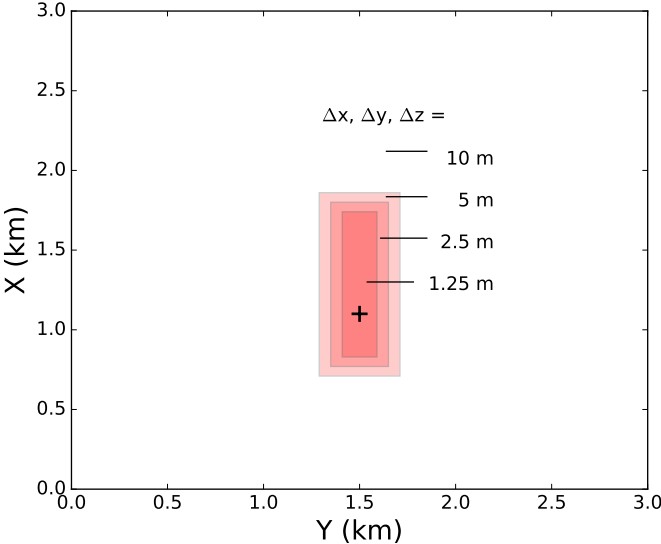

**Figure 2.** Schematic of the SOWFA mesh, with model resolutions labeled at their respective locations in the domain. Turbine location is denoted by the plus sign.

m s$^{-1}$) and below the rated wind speed (12 m s$^{-1}$), a range conducive for generating wakes and typical for general aviation flight conditions.

Horizontal slices of the horizontal wind speed (Figure 3) show the wake characteristics at the near surface and bottom, center, and top of the rotor disk for both cases. The wake in the neutral case is approximately symmetric throughout the rotor
disk (Figure 3b-d), with the two-lobed profile becoming approximately Gaussian around 3D (Figure 3c). In the stable case, the wake exhibits a more asymmetric structure and veers with altitude (Figure 3f-h) due to ambient background veer, as in other stable simulations of wakes (Lundquist et al., 2015). The horizontal extent and width of the wake also differ with stability. The stable conditions enable the wake to maintain a narrower structure and longer downwind presence, as opposed to the neutral conditions that form a comparatively more diffuse and shorter wake. The noise visible in both cases past 8D is due to a
combination of the coarser grid resolution and high sampling resolution in that region of the domain.

## 2.2   Data Analysis

We use the simulation data to quantify potential roll hazards on hypothetical general aviation aircraft transecting the turbulent wake. The Cessna 172, a common general aviation aircraft, has a wingspan of 10 m, a planform area of 16 m$^2$, and an aspect ratio of 7 (Cessna Aircraft Company, 2004). We represent the aircraft in the LES data as a 10-m line. The 1.25-m resolution
of the data allows the line to be divided into eight segments, four points diverging from center to represent the two wings. We then use the LES wind vectors observed at each point on the aircraft to make calculations of the roll hazard metrics: rolling moment and rolling moment coefficient.

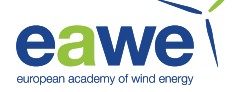
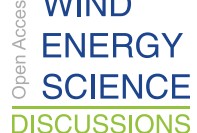


**Figure 3.** Contoured horizontal cross sections of horizontal wind speeds (m s$^{-1}$) comparing the neutral (a–d) and stable (e–h) cases at the near surface (1 m; a, e), rotor bottom (40 m; b, f), rotor center (80 m; c, g), and rotor top (120 m; d, h).



The rolling moment is computed from the lift distribution across the aircraft. At each of the eight points on the aircraft, we sample the oncoming wind velocity vector nonparallel to the wing line. We assume a typical landing flight speed of a general aviation aircraft of 35 m s$^{-1}$; the sampled wind vectors are relative to the movement of the aircraft. Thus, the total velocity vector, or true airspeed, is calculated at each segment: the nonparallel component of the wind vector impinging on the wing,

added to the flight velocity. The elevation angle of each motion-relative total velocity vector forms an angle of attack $\alpha$ along the wing

$$\alpha \equiv \tan^{-1}\frac{w}{V} \tag{1}$$

where $w$ is the instantaneous component of linear velocity in z, and $V$ is the true instantaneous airspeed. From each angle of attack, a coefficient of lift $C_l$ is derived – eight in total along the aircraft

$$C_l = 2\pi\alpha + C_{l,0} = C_{l,b} + C_l' \tag{2}$$

where $C_{l,0}$ is the lift coefficient at zero angle of attack, $C_{l,b}$ is the base coefficient of lift, $C_l'$ is the lift perturbation, and $2\pi\alpha$ is the chosen $C_l$–$\alpha$ curve provided by thin airfoil theory, an accurate idealization for most airfoils at small angles of attack.

The dimensional lift perturbation $L_i'$ of each segment of the aircraft is then derived from the coefficient of lift

$$L_i' = 0.5C_l'\rho|\boldsymbol{V}|^2A \tag{3}$$

where $A$ is the planform area of the segment, and $\rho$ is the standard sea-level atmosphere density of 1.225 kg m$^{-3}$.

To calculate the rolling moment $M_{roll}$, we sum the eight lift values across the length of the wing

$$M_{roll} = \sum_{i=1}^{N} L_i'r_i \tag{4}$$

where $N$ is the total number of points along the aircraft, and $r_i$ is the distance from the center of the aircraft, which is positive (negative) to the left (right) of center from the perspective of the aircraft.

Finally, the rolling moment coefficient $C_{roll}$ is calculated by normalizing the rolling moment by the size and shape of the plane

$$C_{roll} = \frac{2M_{roll}}{\rho|\boldsymbol{V}|^2S\beta} \tag{5}$$

where $|\boldsymbol{V}|$ is the true airspeed averaged across the eight segments, $S$ is the total planform area of the aircraft, and $\beta$ is the aspect ratio of the aircraft, which we choose to be 7 to align with the general aviation Cessna 172 detailed earlier.

The representation of an aircraft as a straight line within the LES data requires several assumptions. For simplicity, we assume a rectangular wing with some base lift equal to the aircraft weight, a result of defining a finite $C_{l,0}$ and some constant aircraft angle of attack. We then only consider a lift perturbation $C_l'$ from this base lift, as the remaining lift distribution is equal on both sides of the wing, contributing no rolling moment. We also note that our calculations assume that the aircraft and the flow field do not interact beyond the first-order effects on the aircraft and thus do not account for any aircraft motion response



due to varying forces. A true depiction of aircraft motion would require a 6-degree-of-freedom aircraft model and solver, more complicated than this study warrants.

We define 540 total flight tracks through the LES data to sample the wind vectors and make the above calculations. These flights are conducted with two different orientations: down-wake transects and cross-wake transects. The down-wake group,
a 10-by-10 array of aircraft separated by 15 grid points (∼18.5 m) in y and z is centered over the wake to define the initial position of each flight path. We march this array of aircraft downwind from the turbine through the x-extent of the domain. The cross-wake group is comprised of a 44-by-10 array of aircraft separated by 15 grid points (∼18.5 m) in x and z initially positioned at y = −2D, the furthest west point of the domain. We march this array of aircraft through the y-extent of the domain, perpendicular to the wake. The cross-wake aircraft descend at a 3° angle to emulate a typical landing pattern. At each
advancing increment in the transect (682 points down-wake, 247 points cross-wake), the rolling moment (Eq. 4) and rolling moment coefficient (Eq. 5) are calculated for all 540 aircraft, yielding 176,880 instances for each roll hazard calculation. We define one additional down-wake flight track far outside of the wake (y = −1.9D, z = 100 m) to serve as a constant, wake-free flight path for comparison. The aircraft do not interact with each other nor modify the flow field as they transect the wake. All calculations are made during a single one-second time step where the aircraft fly across a frozen domain. This process is then
repeated for each of the 100 time steps available, yielding 17,688,000 total hazard calculations.

## 3   Results

We first analyze the turbine wake impacts on a hypothetical general aviation aircraft by comparing roll hazard calculations of a sample flight track outside of the wake to those within the wake. Rolling moment coefficient (Eq. 5) is calculated for an aircraft inside and outside the wake for the neutral (Figure 4b) and stable (Figure 4d) cases to categorize each moment as a
"low", "medium", or "high" hazard as used in the Mulinazzi and Zheng (2014) and Wang et al. (2015) assessments of wake roll hazards. These assessment criteria are based on the maximum rolling moment that the aileron on a typical aircraft can generate to counteract a moment induced by the wake field (Wang et al., 2015). A wake-induced rolling moment coefficient $|C_{roll}|$ above 0.28 is considered a high hazard, between 0.1 and 0.28 a medium hazard, and below 0.1 a low hazard. As expected, in both stability conditions, the aircraft inside the wake experiences higher values of rolling moment coefficient, indicating increased
turbulence of the turbine wake. However, rolling moment coefficients experienced in both stability cases for the aircraft inside the wake remain within the "low" hazard threshold of $|C_{roll}| < 0.1$.

While these sample aircraft flight tracks in Figure (4b, d) show an example for one location in the wake, the LES data set allows us to calculate more instances of roll hazards throughout the entire wake. More conclusive results can be seen by assessing the 540 flight paths down and across the wake spanning all 100 s of data. We summarize occurrences of roll
hazard calculations via histograms of neutral (Figure 5a, b) and stable (Figure 5c, d) cases and down-wake (Figure 5a, c) and cross-wake (Figure 5b, d) transects as a function of the downwind distance from the turbine.

In all transects and stabilities, over 99.99% of all calculations exist within the low hazard threshold. Table 1 lists detailed counts of instances classified as "medium" hazards ($0.1 < |C_{roll}| < 0.28$). The stable cross-wake configuration has the greatest

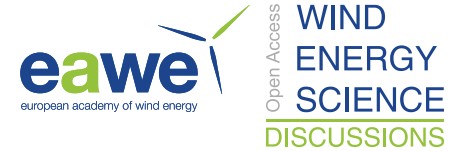

**Table 1.** Summary of roll hazards

| Stability | Transect | $N_{Total}$ | $N_{High}$ | $N_{Medium}$ | Percent Medium |
|-----------|----------|-------------|------------|--------------|----------------|
| Neutral | Down-wake | 6,820,000 | 0 | 74 | 0.001% |
|  | Cross-wake | 10,868,000 | 0 | 72 | 0.0007% |
| Stable | Down-wake | 6,820,000 | 0 | 76 | 0.001% |
|  | Cross-wake | 10,868,000 | 0 | 188 | 0.002% |

fraction of "medium" hazards at 188 out of 10,868,000. Across all cases, no moments reach the "high" hazard threshold. Further, the vast majority of calculated roll hazards are even smaller than the maximum low hazard limit ($|C_{roll}| < 0.1$) and are contained within $|C_{roll}| < 0.02$ (Figure 5). The decreasing frequency of rolling moment coefficients $|C_{roll}|$ greater than 0.02 beyond 8D suggests that roll hazards decrease with downstream distance at this point (Figure 5). While we hypothesized

the stable case would generate more hazardous rolling moment coefficients, we find similar fractions of "medium" hazards in the stable conditions as in neutral (Figure 5). Even though stable conditions enable the wake to persist longer downwind than in convective conditions, they do not increase turbulence that would pose hazards for general aviation aircraft.

The largest roll hazards, as indicated by $C_{roll}$, occur most frequently about 5D downwind from the turbine (Figure 5). The three highest roll hazards in the neutral, down-wake case ($C_{roll} = 0.128, 0.124, 0.123$) all occur in a sequential line downwind

near 6.5D in the top-left quadrant of the rotor disk as looking downwind (black circles in Figure 4a). These rolling moment coefficients are all positive, indicating clockwise rotation about the aircraft's longitudinal axis when looking downwind. In the neutral, cross-wake case, the three highest rolling moments ($C_{roll} = 0.127, 0.124, 0.124$) occur between 3D and 4.5D in the top-right quadrant of the rotor disk (empty circles in Figure 4a). For the stable, down-wake case, the highest two roll hazards ($C_{roll} = -0.121, -0.12$) occur at 3.5D downwind, in the top-left quadrant of the rotor (Figure 4c). The next highest hazard

($C_{roll} = -0.119$) occurs 2.5D downwind, differing from the prior hazards by residing in the bottom-left quadrant of the rotor (Figure 4c). The largest hazards across all cases occur in the stable, cross-wake case at $C_{roll} = 0.14, 0.14, 0.136$, with the first two in the bottom-right quadrant of the rotor disk and the third in the top-left (Figure 4c). All of these peak hazards are located in the high-shear zone at the edge of the wake, which suggests that the rolling moment is more influenced by horizontal shear in the flow than by wake rotation.

**4   Discussion and Conclusion**

As wind energy development increases in the vicinity of general aviation airports, concerns for turbine wake-induced roll hazards on aircraft grow. Of particular concern is the rolling moment, the aerodynamic force applied at a distance from an aircraft's center of mass that causes the aircraft to undergo angular acceleration about its roll axis. Using LES of stable and neutral flow past a utility-scale wind turbine, we quantify the roll hazards experienced by general aviation aircraft transecting

the wake.

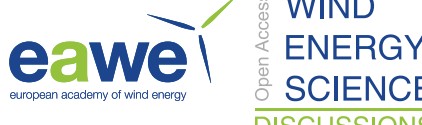

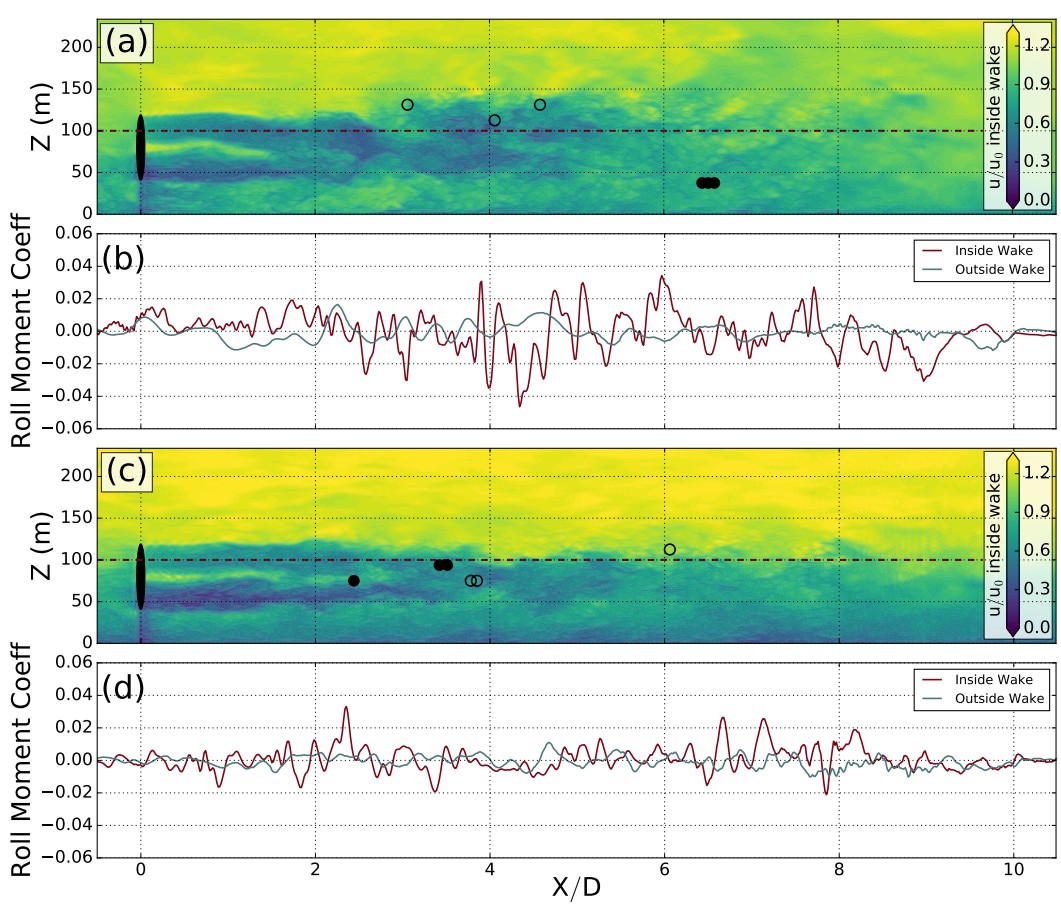

**Figure 4.** Comparison of rolling moment coefficient (b, d) between the flight path inside (red) and outside (blue) of the turbine wake in the neutral (a, b) and stable (c, d) cases. The vertical cross section of the downwind component of the wind velocity u, scaled by inflow wind speed $u_0$, 7.4 and 9.4 m s$^{-1}$ in neutral and stable conditions, respectively, at the turbine location (y = 0; a, c) is contoured for reference, over which the height of the flight paths (z = 100 m) is traced in a red dashed line. The dots represent the downwind and vertical location of the three highest rolling moment coefficients experienced by the down-wake (black) and cross-wake (empty) array of aircraft.



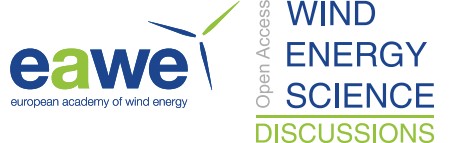

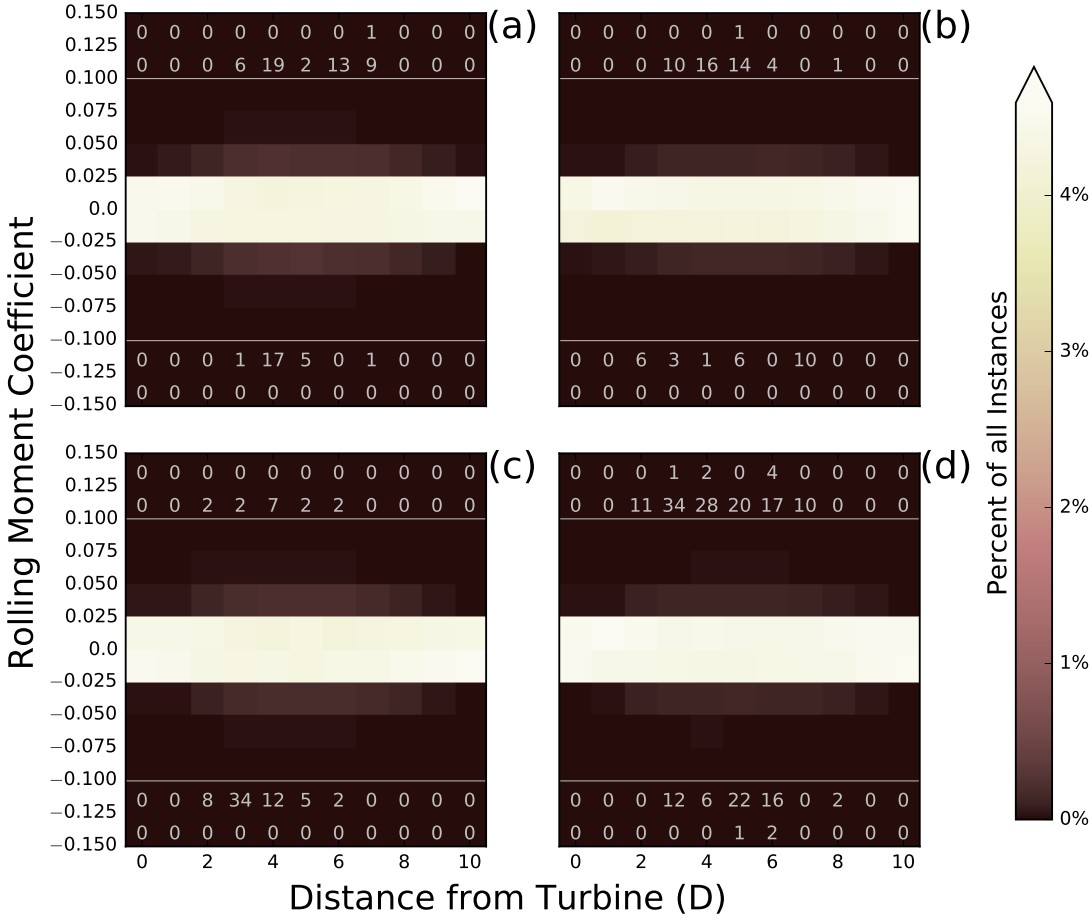

**Figure 5.** Two-dimensional histograms of all calculations of rolling moment coefficient experienced by all aircraft in the neutral (a, b) and stable (c, d) cases for down-wake (a, c) and cross-wake (b, d) transects as a function of the downwind distance from the turbine (in rotor diameters $D$). "Non-low" hazards ($|C_{roll}| > 0.1$) are explicitly totalled in their respective bins.





We represent a typical general aviation aircraft in the LES data as a 10-m line comprised of eight segments that are 1.25 m in length. At each point along the aircraft, we sample the LES wind vectors to calculate rolling moment and the rolling moment coefficient. We define 540 flight tracks to march through the wake to make the roll hazard calculations. The flight tracks have down-wake and cross-wake orientations and extend through the entire downwind 682 points and 247 points, respectively,

yielding 176,880 calculations of a roll hazard for each stability case for each time step of data.

The rolling moment coefficient serves as our primary roll hazard metric, as it normalizes the rolling moment on the aircraft by the aircraft size, shape, and airspeed to produce a standard hazard index. This index has been classified by past studies into thresholds for low, medium, and high roll hazards. When comparing a sample flight path outside of the wake's influence to one within the wake center, we find that an aircraft flying within the wake experiences higher rolling moment coefficient than an

aircraft flying in a wake-free environment. However, analysis of all 540 flight paths through the wake reveal that over 99.99% of hazard calculations remained within the low criterion in both neutrally stratified and stably stratified conditions across 100 s of data. In neutral conditions, the largest of these low hazards are classified as "medium" hazards and exist 5D downwind of the turbine in the upper-left portion of the rotor disk. The highest hazards in the stable case also remained within the medium threshold and are located in two separate regions of the wake: approximately 5D downwind in the upper-left quadrant of the

rotor as in the neutral case, and 8D downwind in the bottom-left quadrant of the rotor.

Our calculated roll hazards differ from those in Mulinazzi and Zheng (2014), who used a helical vortex model scaled up from wind tunnel measurements to find significant roll hazards far downwind from a wind turbine. Conversely, our results using an actuator-line representation of a wind turbine in the SOWFA LES model never surpass the low roll hazard threshold. Rather, our results agree with those of Wang et al. (2015, 2017), whose calculations are computed from the Beddoes circulation formula

(Madsen and Rasmussen, 2004), as opposed to the wind tunnel extrapolation in Mulinazzi and Zheng (2014). Past successes in turbine wake modeling using LES actuator-line methods can in part validate our results (e.g., Churchfield et al. (2012); Nilsson et al. (2015); Jha et al. (2015)).

This study presents a simple method for quantifying turbine wake-induced roll hazards on general aviation aircraft and is constrained by the assumptions made to represent an aircraft within LES data of a single turbine wake. While we have shown

that individual wakes in neutrally stratified and stably stratified conditions are unlikely to pose hazards to general aviation aircraft, interactions between wakes could also be explored. Future simulations of wind plants that allow for wake interaction (as in Vanderwende et al. (2016)) could be useful. Our conclusions are drawn from rolling moment calculations, though additional calculations of yawing and pitching moments could also be beneficial. Our results may also be supported by field tests, in which tethersondes (Lundquist and Bariteau, 2015) or unmanned aircraft vehicles (Kocer et al., 2012; Lawrence and

Balsley, 2013; Båserud et al., 2014) would fly through the wake of a real utility-scale turbine during variable conditions to directly measure roll hazards.





*Code and data availability.* Model output and code from this study is stored on the University of Colorado's PetaLibrary and is available from the authors upon request.

*Competing interests.* The authors declare that they have no conflict of interest.

*Acknowledgements.* This work was supported by a seed grant from Renewable and Sustainable Energy Institute with cooperation from
5  NREL. NREL is a national laboratory of the U.S. Department of Energy, Office of Energy Efficiency and Renewable Energy, operated by the Alliance for Sustainable Energy, LLC. JMT was partially supported by an NSF Graduate Research Fellowship under grant number 1144083. Roll hazard calculations were conducted using the Extreme Science and Engineering Discovery Environment (XSEDE), which is supported by National Science Foundation grant number ACI1053575. JKL's effort was supported by an agreement with NREL under APUP UGA-0-41026-65. The U.S. Government retains and the publisher, by accepting the article for publication, acknowledges that the U.S. Government
10  retains a nonexclusive, paid-up, irrevocable, worldwide license to publish or reproduce the published form of this work, or allow others to do so, for U.S. Government purposes.



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
