# Peer review of "Do Wind Turbines Pose Roll Hazards to Light Aircraft?"

_Wind Energy Science, 2018_

## Referee Comment (RC1) · Anonymous Referee #1 · 16 Jul 2018

In the light of close neighbourhood of wind farms and small airport for general aviation in the USA this manuscript examines the question if the wake behind a wind turbine can pose a hazard to light aircraft. The study is based on large-eddy simulations (LES) of neutrally and stably stratified turbulent boundary layers in which the wake flow behind an actuator-line parameterised wind turbine is computed. As a measure of potentially hazardous wake encounters of a light airplane its rolling moment coefficient is computed for many sets of flight tracks in down-wake and cross-wake directions in the LES boundary layer. The aircraft (a/c) solely consists of a wing which is modelled as a line segmented in eight stripes. Both, the wind turbine and the light aircraft are chosen as typical representatives of their kind in the USA.

The paper is well written and good to understand for readers with background in either

wind energy or aviation and it should be of high interest for both groups. Since the study adds to the ongoing and contradictory discussion if or if not light a/c are in danger in wind turbine wakes it is of utmost importance that the methodology and assumptions are properly explained and justified.

I recommend publication of the paper after the authors have addressed my comments and questions.

My main concern addresses the simplification of the forces acting on the a/c, p. 7/8: The authors do not account for the a/c motion response (p.7 line 29). However, the swirl in the wake has lateral and vertical flow components. Hence, when the a/c has started to roll not only w changes the angle of attack (eq. 1) but also the lateral (wing-parallel) wind component v, which acts on the wing, yielding a higher roll rate, bank angle and rolling moment coefficient. So, is it justifiable to neglect that response? In the view of my argument (if it is true) the $C\_roll$ values obtained and discussed might be an underestimate of the total roll effect and therefore critical.

Classification of the rolling moment coefficient on p. 8, lines 22-23: The classification of an a/c roll as a "low", "medium", or "high" hazard very strongly depends of the flight altitude of the a/c above ground. The same $C\_roll$ value being classified as "medium" or "low" when the a/c is flying high above ground might be classified as "high" when the a/c is close to the ground as is the case here. So, I wonder for which flight altitude the thresholds mentioned in the paper are valid. Are these the correct values for a/c flying at wind turbine rotor height ? In the light of that question it is very valuable that the authors present all the $C\_roll$ values and state on p.9 lines 2-3: ...vast majority ... are contained within $|C\_roll| < 0.02...$". This result together with the ambiguity of the classification should be discussed / interpreted in Chapter 4.

Shear versus wake rotation on p. 9 line 18-19: Since the swirl of the wake (the wake rotation) is strongest where also the shear is large (at the edge of the wake) it is hard to decide if the rolling moment is mainly caused by uncoherent turbulence due to shear

or by the more coherent motion of the swirl. The data are available from the LES to do that discrimination, although I would understand that this might be beyond the scope of the paper.

The fleet of a/c encountering the wake in down-wake (cross-wake) trajectories counts 10x10 (44x10), a/c, leaving space between a/c pairs, p. 8, lines 4-8: If my understanding is correct, then not the entire domain is searched for C_roll and there is a chance that maybe the most hazardous parts in the wake are not found because they are just between two a/c. Wouldn't it be better therefore to place (virtual) a/c at _each_ grid-point of the domain (with overlapping wings) to cover the entire wake ? (This would also increase the already impressive sample size drastically.)

Some other points which I came across:

P. 5 line 3: I guess the instantaneous horizontal wind field is plotted.

Fig. 3 on p. 6: The authors mention the numerical noise beyond 8 D downstream which appears as an organised "wavy" structure of the horizontal wind. This structure can also be seen (with a somewhat weaker signal though) at other, more relevant locations in the plotted horizontal cross-sections (e.g. between at 3 and 7D of Fig. 3b and f, close to the lateral boundaries at y=+/- 1.5D. Does this indicate some numerical instability due to the changing grid resolution laterally ? And if so, does it have an impact on the results ?

P. 7 line 1: better: ". . . across the aircraft's wingspan."

p. 7 line 8: why "linear" velocity ?

p. 9 line 10: "top-left" correct here ?

p.12 lines 13-15: I cannot match the statement (positions) here with the dots in Fig. 4. Please explain or reformulate.

---

## Referee Comment (RC2) · Anonymous Referee #2 · 20 Jul 2018

This reviewer is familiar with classical aerodynamics and flight mechanics, but less educated in the fluid mechanics and the industry development of wind turbines. Therefore I find the introduction useful as written, since it well describes the relevance of the topic and it provides the readers an overview of the wake computing methodologies and literature in the field.

Overall I am supportive of the publication of this material, but would like the authors to be more explicit on certain findings. I therefore suggest that the manuscript be further revised, and would recommend this manuscript be conditionally accepted at this time. I would like the authors to provide some additional insight or share more of their experience in the following:

1. Do the authors have evidence that the helical tip vortices generated by the wind tur-

bine can be meaningfully captured at the source, as well as its downstream evolution, in the computational methodology employed in the paper?

2. Do the authors have evidence that these helical vortices either dissipate quickly (i.e., physically instead of numerically) or is not a flow feature that contributes to worst case encounters?

Other Specific comments: Because this reviewer is not very familiar with the wind turbine industry, I do not know the significance of the phrase "utility-scale wind turbine". Is that supposed to represent the upper range in the spectrum of wind turbine power extraction (therefore the upper limit of the wake generator)? If so, it would be helpful to point that for the general readers.

Related comment: just because GE 1.5-MW SLE is a common model, it does not necessarily bound the risk. This reviewer uses the word "risk" in the context of the FAA Safety Management System (SMS), in that risk is a combination of likelihood and severity of a hazard. If the turbine being simulated were a common model (at least stated so in the manuscript), it is at least meaningful in a likelihood/frequency perspective. However, it may not be as meaningful in a potential severity perspective. It would thus be useful to provide the additional insight in terms of where this wind turbine resides in the spectrum of characteristics like physical size, power extraction level, etc., in wind turbine industry. As an example, in the area of aircraft wake turbulence, the most common aircraft in commercial aviation is Airbus A320 (a single isle commercial transport). But that airplane, although most common, does not represent the upper bound of the wake turbulence severity. Airbus A380 for example, is seven times heavier than that of the A320 and generates far more significant wake turbulence.

This reviewer supports the choice of the actuator-line approach, as it is claimed in the manuscript that this representation of the wind turbine can capture tip vortices. It would be useful for the authors to more explicitly illustrate that the computed flow field did indeed capture the tip vortices (and the associated initial circulation is considered

reasonable).

This reviewer supports the inclusion of the neutral atmospheric condition in the simulation. It is likely to contribute to near worse conditions for turbine wake evolution. However, the decay of tip vortices is also influenced by turbulence level, and it would be useful to comment on the turbulence level both in the incoming flow as well representative locations in the wake. EDR is often preferred (only because it is more often used in aircraft wake turbulence), but TKE would be meaningful as well.

This reviver is not familiar with SOWFA, nor OpenFOAM. It would be useful to comment on the Reynolds number associated with a real life utility class wind turbine vs. what is used in the simulation. It would be also insightful to comment on how the incoming boundary layer flow is handled (or at least reference it if it is too lengthy). It would be preferred if the results in 8D are less impacted by numerical noise. However, if the CFD results are considered realistic enough, encounter scenarios in regions prior to 8D should help argue that conservative estimates have been made within the current framework.

Equation 5 appears to be dimensionally inconsistent.

At least in terms of the aircraft wake vortex community, there is no universal acceptance on a severity criteria (a web based, not most authoritative but easily accessible reference would be the power point material from "van der Geest, WakeNet3 Europe, Feb 2012 – "Wake Vortex Severity Criteria, the Search for a Single Metric""). However, it is commonly accepted that roll moment coefficient (a static severity matric) is a reasonable predictor of the aircraft response. However, it is also argued that roll moment coefficient is more relevant when it comes to Large aircraft (when the aircraft roll response is slow in the presence of a wake) and most applicable to Heavy category aircraft. As the authors have correctly pointed out, severity is better described in terms of aircraft response. However, pilot reactions and perceptions are perhaps, at the end of the day, form the most relevant hazard definition. And pilot perception involves a

complicated and subjective set of criteria that include the aircraft energy state, altitude and performance. This reviewer is not expecting the present authors to resolve a topic that the wake vortex community has not been able to resolve for 40 years in the US (and 20 years in Europe). Instead, it should be recognized that roll moment coefficient has certain limitation. However, if roll moment coefficient were used in a relative sense against a recognized safe baseline, even though the response is not characterized, it leads to a better argument. It is for this reason that this review does not believe the roll moment coefficient based boundaries developed by Mulizanni and Zheng (2014) are very meaningful. In addition, roll moment coefficient has various levels of approximations in its formulation and the computed value can differ by a factor of close to two for the same flow input. It is believed that the formulation used by the authors is to represent the wing of a typical GA aircraft as a rectangular lifting surface, and this treatment is conceptually consistent with the formulation used in developing a set of baseline roll moment coefficient exposures in wake turbulence (see Fig 5 of AIAA-2016-3434, and its reference 10). If the roll moment coefficient exposure in wind turbine wakes are not anywhere close to the wake turbulence baseline exposure, then an argument can be made that the exposure from turbine wakes are acceptably safe, or just as safe or safer than the exposure due to the ICAO wake turbulence separation, however aircraft respond to those levels of roll moment coefficients.

This review is very interested in the following feedback from the authors: It is not clear that all of the features that potentially pose roll hazards are properly captured. I am particularly interested in knowing if the authors have confidence in their computed field in terms of the proper capturing, as well as the proper decay of helical tip vortices. Mulinazzi and Zheng (2014) used the near field turbine tip vortex data from wind tunnel measurements and assumed the scaling relationship that governs the decay of vortices to be the same as that of the aircraft wake vortices. The decay of the vortices is driven by the ambient turbulence in their formulation, which may not be as realistic since conceptually the vortex decay should be most influenced by the turbulence field surrounding the tip vortices (or in this case, turbulence level in the wake of the wind

turbine itself). The decay of the tip vortices in Mulinazzi and Zheng (2014) may therefore represent a conservative scenario. The encounter scenario used in Mulinazzi and Zheng (2014) is then for the small aircraft to hit those highly coherent swirling structures that have sharp velocity gradients in the transverse direction relative to the wingspan of the aircraft. This treatment is considered reasonable and conservative (not necessarily a bad thing for commenting on safety), especially if tip vortices have been shown in the wind turbine literature to persist longer than other velocity deficit or turbulent features. However, once again, the computed tip vortex decay may be too conservative in Mulinazzi and Zheng (2014), since the source of wake decay is taken to be the ambient turbulence instead of the turbulence within the velocity deficit region of the turbine wake. If the LES computation by the authors were shown to be capable of capturing the initial generation of the blade tip vortices with the correct range of strength/circulation, and the computational technique is capable of preserving vorticity without artificial numerical dissipation/diffusion of the vortex, and the encounter scenario involves entering the properly computed / surviving tip vortex, and the conclusion is still that roll moment coefficient is considered low relative to a safe baseline, then it would completely satisfy the current reviewer. This reviewer would like to have some assurance that the results are truly due to all of the possibly relevant flow features being properly captured, and the results are not biased by the computed flow field that cannot capture the critical features that may be important for this specific problem. The tip vortex feature may not be as important in traditional wind turbine wake applications such as siting optimization, but it is considered potentially more important than other features in terms of roll upset. If the numerical scheme and associated modeling of the turbine cannot meaningfully duplicate the tip vortex flow field, then this review would suggest that the wording in the conclusion be modified along the phrases used in Wang, White and Barakos (2017). The aforementioned reference essentially pointed out the flow features their model and LIDAR measurements can reveal, and estimated the risk is only based on those features that their flow field data can resolve.

This review once again, thank the authors for the effort to advance the knowledge in

this field.

---

## Referee Comment (RC3) · Anonymous Referee #3 · 24 Jul 2018

This manuscript attempts addressing the issue about flight hazards for airplanes flying in the wake of utility-scale wind turbines. The investigation is carried out through LES simulations and the results are interesting, which might deserve to be disseminated. Please find below some comments.

Comments:

1. P1, L1-4: The first part of the abstract should be more focused and detailed on the motivation, procedure and results. This first four lines sound more appropriate for an introduction; indeed, similar information is reported in Sect. 1.

2. P1, L7: You could add that you deem stable and neutral conditions more critical than convective conditions due to the faster wake recovery. Therefore, you did not

performed simulations under convective conditions.

3. P2, L9 : Why do you only consider roll and not, for instance, pitch moment? Maybe there are safety standards in general aviation that I am not aware of. In that case, please provide related references. I think that a non-symmetric velocity field induced by the wind turbine wake can also affect pitch and yaw of the airplane, which might be a risky situation leading to a premature wing stall. Please comment on this and, eventually, clarify.

4. P2, L11-12: "The rolling moment is the aerodynamic force applied", a moment cannot be a force. Maybe rephrase it saying that the roll moment is the result of the lift distribution over the wing span.

5. Eq. 5 is inconsistent. \beta should be the wing span, not the aspect ratio. I hope this being only a typo and not jeopardizing your analysis. Please cross-check your data analysis as well.

6. P8, L5-7: "a 10-by-10 array of aircraft", this description is not clear. I think It will be better to talk about aircraft paths rather than aircraft arrays. Please try to rephrase it.

---

## Author Comment (AC1) · 13 Sep 2018

All reviewer comments appear in regular text below, while authors' responses appear in purple text. Line numbers referenced in the authors' responses refer to the revised document.

**Response to Anonymous Referee #1**

In the light of close neighbourhood of wind farms and small airport for general aviation in the USA this manuscript examines the question if the wake behind a wind turbine can pose a hazard to light aircraft. The study is based on large-eddy simulations (LES) of neutrally and stably stratified turbulent boundary layers in which the wake flow behind an actuator-line parameterised wind turbine is computed. As a measure of potentially hazardous wake encounters of a light airplane its rolling moment coefficient is computed for many sets of flight tracks in down-wake and cross-wake directions in the LES boundary layer. The aircraft (a/c) solely consists of a wing which is modelled as a line segmented in eight stripes. Both, the wind turbine and the light aircraft are chosen as typical representatives of their kind in the USA.

The paper is well written and good to understand for readers with background in either wind energy or aviation and it should be of high interest for both groups. Since the study adds to the ongoing and contradictory discussion if or if not light a/c are in danger in wind turbine wakes it is of utmost importance that the methodology and assumptions are properly explained and justified.

I recommend publication of the paper after the authors have addressed my comments and questions.

1. My main concern addresses the simplification of the forces acting on the a/c, p. 7/8: The authors do not account for the a/c motion response (p.7 line 29). However, the swirl in the wake has lateral and vertical flow components. Hence, when the a/c has started to roll not only w changes the angle of attack (eq. 1) but also the lateral (wingparallel) wind component v, which acts on the wing, yielding a higher roll rate, bank angle and rolling moment coefficient. So, is it justifiable to neglect that response? In the view of my argument (if it is true) the C_roll values obtained and discussed might be an underestimate of the total roll effect and therefore critical.

Thank you for your thoughtful point. Because the aircraft response would introduce a cascade of events that change in time, we don't see a way to include this cascade of events on the aircraft. To acknowledge this uncertainty, we added justification for ignoring motion response in page 8 lines 10-14:

"We recognize that by not accounting for the aircraft motion response, these calculations may omit cases when the changing wind components on the wing could result in higher rolling

moments. However, such an additional series of calculations would introduce uncertainty because of the role of pilot response, and we argue that the wake roll hazards encountered (to be explained in Section 3) would allow a pilot to quickly correct against wake turbulence-induced roll instead of allowing roll to escalate."

2. Classification of the rolling moment coefficient on p. 8, lines 22-23: The classification of an a/c roll as a "low", "medium", or "high" hazard very strongly depends of the flight altitude of the a/c above ground. The same C_roll value being classified as "medium" or "low" when the a/c is flying high above ground might be classified as "high" when the a/c is close to the ground as is the case here. So, I wonder for which flight altitude the thresholds mentioned in the paper are valid. Are these the correct values for a/c flying at wind turbine rotor height ? In the light of that question it is very valuable that the authors present all the C_roll values and state on p.9 lines 2-3: . . .vast majority . . . are contained within |C_roll| < 0.02. . .". This result together with the ambiguity of the classification should be discussed / interpreted in Chapter 4.

The classifications of the rolling moment coefficients are here chosen to be consistent with those in the previous literature, specifically Mulinazzi et al. (2014), who also considered aircraft flying near ground level. We have changed the text on page 7 lines 22-23 to emphasize that air density is included in the rolling moment calculation, which accounts for altitude:

"where A is the planform area of the segment, and \rho is the standard near sea-level atmosphere density of 1.225 kg m^-3 to reflect the altitude of interest for wake-transecting aircraft."

3. Shear versus wake rotation on p. 9 line 18-19: Since the swirl of the wake (the wake rotation) is strongest where also the shear is large (at the edge of the wake) it is hard to decide if the rolling moment is mainly caused by uncoherent turbulence due to shear or by the more coherent motion of the swirl. The data are available from the LES to do that discrimination, although I would understand that this might be beyond the scope of the paper.

Thank you for raising an interesting point. We agree with the reviewer that discriminating between coherent and incoherent turbulence effects is beyond the scope of the paper. We have hypothesized on page 10 lines 2-5 that the location of the highest hazards 3-7D downwind from the turbine (and not closer to the turbine, e.g. 0.5-1D, where swirl would be strongest) suggests greater influence from incoherent shear-induced turbulence:

"All of these peak hazards are located in the high-shear zone at the edge of the wake between 3 and 7D downwind from the turbine (and not closer to the turbine, e.g. 0.5-1D), which suggests that the rolling moment is more influenced by horizontal shear in the flow than by wake rotation."

4. The fleet of a/c encountering the wake in down-wake (cross-wake) trajectories counts 10x10 (44x10), a/c, leaving space between a/c pairs, p. 8, lines 4-8: If my understanding

is correct, then not the entire domain is searched for C_roll and there is a chance that maybe the most hazardous parts in the wake are not found because they are just between two a/c. Wouldn't it be better therefore to place (virtual) a/c at _each_ gridpoint of the domain (with overlapping wings) to cover the entire wake ? (This would also increase the already impressive sample size drastically.)

We agree that placing a virtual aircraft at each grid point would increase the sample size. Unfortunately, we are limited by computation constraints, and adding calculations at all grid points would roughly increase the computation time over 5 times. We carefully considered the reviewer's suggestion and reran our calculations for a small part of the domain (5D < X < 5.5D, -0.25D < Y < 0.25D, Z = 100m) where the highest rolling moments were found on the initial study. Because we did not find any larger rolling moments even in those regions, we are comfortable stating that overlapping wings are not necessary. As the reviewer noted, we already have an "impressive" sample size.

Some other points which I came across:

5. P. 5 line 3: I guess the instantaneous horizontal wind field is plotted. Fig. 3 on p. 6: The authors mention the numerical noise beyond 8 D downstream which appears as an organised "wavy" structure of the horizontal wind. This structure can also be seen (with a somewhat weaker signal though) at other, more relevant locations in the plotted horizontal cross-sections (e.g. between at 3 and 7D of Fig. 3b and f, close to the lateral boundaries at y=+/- 1.5D. Does this indicate some numerical instability due to the changing grid resolution laterally ? And if so, does it have an impact on the results ?

The numerical noise visible in the far outer edges of the innermost domain is indeed due to the decreasing grid resolution, and this is emphasized in page 5 lines 16-17:

"The noise visible in both cases past 8D is due to a combination of the coarser grid resolution and high sampling resolution in that region of the domain."

Further, the computations were performed on a computational grid of variable resolution in which the grid lines were Cartesian oriented. However, the wind direction and turbine orientation were not Cartesian aligned (i.e. the wind came from the southwest rather than the south, and the turbine was pointed directly into this wind). The grid refinement box followed this wind direction to capture the wake. We then sampled the CFD solution onto a uniform resolution mesh aligned with the wind and rotated this sampling mesh to be Cartesian aligned for ease of post-processing and presentation. Yes, a small amount of oscillation in the solution is numerical and caused by the flow seeing a jump in resolution, but a larger component of it is an artifact of the oversampling of the coarse computational mesh regions onto the uniformly fine, rotated sampling mesh. Because, in the computation, these oscillations are much smaller than they appear here, we do not believe they adversely affect the solution. The calculated roll hazards within these regions are very similar to those adjacent, where numerical noise is not present.

6. P. 7 line 1: better: ". . . across the aircraft's wingspan."

Thank you; the text has been amended in page 7 line 8.

7. p. 7 line 8: why "linear" velocity ?

The word "linear" was removed to avoid confusion in page 7 line 15.

8. p. 9 line 10: "top-left" correct here ?

Thank you for catching this typo, page 9 line 22 has been amended to say "bottom-left".

9. p.12 lines 13-15: I cannot match the statement (positions) here with the dots in Fig. 4. Please explain or reformulate.

Thank you, this statement on page 12 lines 18-21 has been updated to read:

"In neutral conditions, the largest of these low hazards are classified as "medium" hazards and exist 6.5D downwind of the turbine in the bottom-left portion of the rotor disk. The highest hazards in the stable case also remained within the medium threshold and are located in two separate regions of the wake: approximately 4D downwind in the bottom-right quadrant of the rotor and 6D downwind in the top-left quadrant of the rotor."

**Response to Anonymous Referee #2**

This reviewer is familiar with classical aerodynamics and flight mechanics, but less educated in the fluid mechanics and the industry development of wind turbines. Therefore I find the introduction useful as written, since it well describes the relevance of the topic and it provides the readers an overview of the wake computing methodologies and literature in the field.

Overall I am supportive of the publication of this material, but would like the authors to be more explicit on certain findings. I therefore suggest that the manuscript be further revised, and would recommend this manuscript be conditionally accepted at this time. I would like the authors to provide some additional insight or share more of their experience in the following:

1. Do the authors have evidence that the helical tip vortices generated by the wind turbine can be meaningfully captured at the source, as well as its downstream evolution, in the computational methodology employed in the paper?

The authors understand the dependence of the actuator line-generated tip vortex on actuator line input parameters. During the work toward this paper, the authors were also engaged in research on this topic that was later published. The citations are given here and included in page 4 lines 21-23:

"The turbine model consists of an actuator-line representation of turbine blades (Sørensen and Shen, 2002), proven to adequately capture the generation and downstream evolution of helical tip vortices (Ivanell et al., 2010; Lignarolo et al., 2015; Toloui et al., 2015; Churchfield et al., 2017; Martínez-Tossas et al., 2017)."

Matthew J. Churchfield, Scott J. Schreck, Luis A. Martínez, Charles Meneveau, and Philippe R. Spalart. "An Advanced Actuator Line Method for Wind Energy Applications and Beyond", 35th Wind Energy Symposium, AIAA SciTech Forum, (AIAA 2017-1998) https://doi.org/10.2514/6.2017-1998

L. A. Martínez-Tossas, M. J. Churchfield, and C. Meneveau, "Optimal smoothing length scale for actuator line models of wind turbine blades based on Gaussian body force distribution." Wind Energy, Vol. 20, Issue 6, pp. 1083—1096, June 2017.

2. Do the authors have evidence that these helical vortices either dissipate quickly (i.e., physically instead of numerically) or is not a flow feature that contributes to worst case encounters?

Tip vortices generated by wind turbine blades are known to break down relatively quickly. Compared to aircraft trailing vortices, which can persist for kilometers, wind turbine tip vortices are subject to atmospheric turbulence, are arranged in a tight helical pattern, and are situated in the shear layer of the wake, all of which lead to vortex structure instability and eventual breakdown. Tip vortex spirals from wind turbines in atmospheric turbulence quickly begin to "leap frog" in which one spiral mutually induces motion on the next spiral and they wrap around

each other. Complete breakdown quickly follows leapfrogging. The tip vortices are usually non-apparent within 1-2 rotor diameters downstream under normal atmospheric turbulence levels. Good references on this subject are below, and we have noted them in page 4 lines 21-23:

"The turbine model consists of an actuator-line representation of turbine blades (Sørensen and Shen, 2002), proven to adequately capture the generation and downstream evolution of helical tip vortices (Ivanell et al., 2010; Lignarolo et al., 2015; Toloui et al., 2015; Churchfield et al., 2017; Martínez-Tossas et al., 2017)."

Toloui, Mostafa & Chamorro, Leonardo & Hong, Jiarong. (2015). Detection of tip-vortex signatures behind a 2.5 MW wind turbine. Journal of Wind Engineering and Industrial Aerodynamics. 143. 10.1016/j.jweia.2015.05.001.

Lignarolo, Lorenzo & Ragni, Daniele & Scarano, Fulvio & Ferreira, Carlos & van Bussel, Gerard. (2015). Tip-vortex instability and turbulent mixing in wind-turbine wakes. Journal of Fluid Mechanics. 781. 467-493. 10.1017/jfm.2015.470.

Ivanell, Stefan & Mikkelsen, Robert & Sørensen, Jens & Henningson, Dan, Stability analysis of the tip vortices of a wind turbine. Wind Energy, Vol. 13, Issue 8, pp. 705—715, Nov. 2010.

3. Other Specific comments: Because this reviewer is not very familiar with the wind turbine industry, I do not know the significance of the phrase "utility-scale wind turbine". Is that supposed to represent the upper range in the spectrum of wind turbine power extraction (therefore the upper limit of the wake generator)? If so, it would be helpful to point that for the general readers.

The phrase "utility-scale wind turbine" typically means that the turbine exceeds 1000 kilowatts in size and would be deployed in standard industrial wind farms (but not necessarily in the upper range of power extraction). Text has been added to Page 4 line 24 to reflect this fact:

"We model the GE 1.5-MW SLE wind turbine (Mendoza et al., 2015), a utility-scale turbine often deployed in standard industrial wind farms."

4. Related comment: just because GE 1.5-MW SLE is a common model, it does not necessarily bound the risk. This reviewer uses the word "risk" in the context of the FAA Safety Management System (SMS), in that risk is a combination of likelihood and severity of a hazard. If the turbine being simulated were a common model (at least stated so in the manuscript), it is at least meaningful in a likelihood/frequency perspective. However, it may not be as meaningful in a potential severity perspective. It would thus be useful to provide the additional insight in terms of where this wind turbine resides in the spectrum of characteristics like physical size, power extraction level, etc., in wind turbine industry. As an example, in the area of aircraft wake turbulence, the most common aircraft in commercial aviation is Airbus A320 (a single isle commercial transport). But that airplane, although most common, does not represent the upper bound of the wake

turbulence severity. Airbus A380 for example, is seven times heavier than that of the A320 and generates far more significant wake turbulence.

The authors agree that the GE 1.5-MW SLE represents the most ubiquitous amount of hazard, not the maximum. However, large eddy simulations were only available for this turbine. In light of the reviewers' comments, we added on page 12 line 33 a suggestion for future work to consider larger turbine types:

"Future simulations of wind plants that allow for wake interaction (as in Vanderwende et al. (2016)) and larger turbine types could be useful."

5. This reviewer supports the choice of the actuator-line approach, as it is claimed in the manuscript that this representation of the wind turbine can capture tip vortices. It would be useful for the authors to more explicitly illustrate that the computed flow field did indeed capture the tip vortices (and the associated initial circulation is considered reasonable).

When we run actuator line calculations, the lift and drag along the line are projected onto the flow field as a volumetric body force using a Gaussian distribution along the line. This yields a thin tube or cloud of body force along the line that is strongest at the line and then exponentially decays with distance from the line. The tunable parameter in the actuator line is the width of this Gaussian distribution. We always perform a tuning procedure in which we run three cases in uniform inflow wind with different Gaussian widths. We then look at how predicted power changes depending upon this Gaussian width. Given the known power, we can then find the Gaussian width that will cause the simulation to predict the desired power. We do this for one wind speed because we have done other tests in the past that show that the single correct or optimal Gaussian width for one wind speed, then holds for the other wind speeds.

This Gaussian width also dictates the width of the core size of the tip vortex. The wider the Gaussian, the wider and more diffuse the tip vortex; however in all cases, given a fixed lift, no matter the Gaussian width, the circulation will be the same. So, no matter what, we begin with the correct circulation, it is just a matter of do we predict a tight enough initial vortex. The "tightness" of the vortex then dictates the downwash distribution along the blade, which affects the predicted loads, and hence power. We therefore consider the optimal, tuned Gaussian width to produce the correct vortex "tightness" because the power is right, meaning the loads along the blade, and hence the downwash is correct. We have experimented with this extensively for both rotating actuator lines and for actuator lines meant to represent fixed wings. Please see

L. A. Martínez-Tossas, M. J. Churchfield, and C. Meneveau, "Optimal smoothing length scale for actuator line models of wind turbine blades based on Gaussian body force distribution." Wind Energy, Vol. 20, Issue 6, pp. 1083—1096, June 2017.

and any more recent work by Martínez-Tossas and Meneveau, with whom we work closely.

6. This reviewer supports the inclusion of the neutral atmospheric condition in the simulation. It is likely to contribute to near worse conditions for turbine wake evolution. However, the decay of tip vortices is also influenced by turbulence level, and it would be useful to comment on the turbulence level both in the incoming flow as well representative locations in the wake. EDR is often preferred (only because it is more often used in aircraft wake turbulence), but TKE would be meaningful as well.

We are pleased the reviewer supports our examination of both neutral and stable atmospheric conditions. Several studies have shown that wakes survive further downwind in stable conditions (with lower ambient TKE or EDR) than in neutral or convective conditions. If the ambient or inflow turbulence was higher, such as in convective simulations, the tip vortices would erode faster. We have specifically added reference to TKE on page 4 lines 34-35 and page 5 lines 1-4:

"Previous observations note turbine wakes tend to diffuse more rapidly in convective conditions because **the high ambient turbulent kinetic energy (TKE)** of the surrounding air **induces mechanical mixing to erode** the wake (Baker and Walker, 1984; Magnusson and Smedman, 1994; Bhaganagar and Debnath, 2014; Mirocha et al., 2015). We thus hypothesize that stable conditions**, with low TKE and low turbulent eddy dissipation rate (EDR) (Bodini et al. 2018),** present a worst-case scenario for general aviation aircraft due to longer-persisting wakes permitted by the reduced ambient **TKE and EDR**. As such, we simulate a neutral case and a stable case."

7. This reviver is not familiar with SOWFA, nor OpenFOAM. It would be useful to comment on the Reynolds number associated with a real life utility class wind turbine vs. what is used in the simulation. It would be also insightful to comment on how the incoming boundary layer flow is handled (or at least reference it if it is too lengthy). It would be preferred if the results in 8D are less impacted by numerical noise. However, if the CFD results are considered realistic enough, encounter scenarios in regions prior to 8D should help argue that conservative estimates have been made within the current framework.

We agree that the numerical noise past 8D is not ideal, but the calculated roll hazards at those points are very similar to those just before 8D.

The Reynolds number of the atmospheric boundary layer is $O(10^8) - O(10^9)$. The Reynolds number on modern utility scale turbine blades is $O(10^6) - O(10^7)$. Direct numerical simulations (DNS) are not possible at these high Reynolds numbers, so we perform large-eddy simulations (LES) in which the larger, energy-containing turbulent scales are directly resolved, and the effect of the smaller unresolved scales is modeled with a subgrid-scale turbulence model.

The incoming boundary layer is computed in a separate "precursor" atmospheric large-eddy simulation using standard practices among the atmospheric LES community. It should also be pointed out that LES was born out of the atmospheric boundary layer community, so these practices have been around and vetted since the 1970s. Basically, the domain is a large atmospheric box extending from the ground up a kilometer or two and 3-5 kilometers

horizontally. The lateral boundaries are periodic. The equations carry a Boussinesq buoyancy term and solve for temperature transport to account for density stratification, which affect turbulence. Surface heating or cooling can be applied. Also, the surface is treated as a rough wall, like the surface of the Earth. The simulation is run for many hours of physical time until a fully developed atmospheric boundary layer is formed. We then sample inflow planes of velocity and potential temperature and feed this into the wind turbine simulation, which uses all the same numerics and physical models. This process is explained further in citation Churchfield et al. (2012) in page 4 line 11.

8. Equation 5 appears to be dimensionally inconsistent.

We thank you for catching this issue and have corrected it. Page 8 lines 1-2 should have said "b is the wingspan of the aircraft, which we set to be 10 m", and we have edited it accordingly. The calculations in the analysis were correct and remain unchanged, but now the misnomer of a variable is fixed.

9. At least in terms of the aircraft wake vortex community, there is no universal acceptance on a severity criteria (a web based, not most authoritative but easily accessible reference would be the power point material from "van der Geest, WakeNet3 Europe, Feb 2012 – "Wake Vortex Severity Criteria, the Search for a Single Metric""). However, it is commonly accepted that roll moment coefficient (a static severity matric) is a reasonable predictor of the aircraft response. However, it is also argued that roll moment coefficient is more relevant when it comes to Large aircraft (when the aircraft roll response is slow in the presence of a wake) and most applicable to Heavy category aircraft. As the authors have correctly pointed out, severity is better described in terms of aircraft response. However, pilot reactions and perceptions are perhaps, at the end of the day, form the most relevant hazard definition. And pilot perception involves a complicated and subjective set of criteria that include the aircraft energy state, altitude and performance. This reviewer is not expecting the present authors to resolve a topic that the wake vortex community has not been able to resolve for 40 years in the US (and 20 years in Europe). Instead, it should be recognized that roll moment coefficient has certain limitation.

We agree with the reviewer that pilot reactions are important, and so we have added a sentence to the conclusion in page 13 lines 3-5 that "Future studies could integrate simulations like these with flight simulators to understand the coupling of the atmospheric behavior to pilot response for an integrated assessment of roll hazard, as in Wang et al. 2015."

10. However, if roll moment coefficient were used in a relative sense against a recognized safe baseline, even though the response is not characterized, it leads to a better argument. It is for this reason that this review does not believe the roll moment coefficient based boundaries developed by Mulizanni and Zheng (2014) are very meaningful. In addition, roll moment coefficient has various levels of approximations in its formulation and the computed value can differ by a factor of close to two for the same flow input. It is

believed that the formulation used by the authors is to represent the wing of a typical GA aircraft as a rectangular lifting surface, and this treatment is conceptually consistent with the formulation used in developing a set of baseline roll moment coefficient exposures in wake turbulence (see Fig 5 of AIAA-2016-3434, and its reference 10). If the roll moment coefficient exposure in wind turbine wakes are not anywhere close to the wake turbulence baseline exposure, then an argument can be made that the exposure from turbine wakes are acceptably safe, or just as safe or safer than the exposure due to the ICAO wake turbulence separation, however aircraft respond to those levels of roll moment coefficients.

We agree with the reviewer that the boundaries developed by Mulinazzi et al. (2014) are not ideal, but we thought it best to be consistent with previous literature.

11. This review is very interested in the following feedback from the authors: It is not clear that all of the features that potentially pose roll hazards are properly captured. I am particularly interested in knowing if the authors have confidence in their computed field in terms of the proper capturing, as well as the proper decay of helical tip vortices.

Please see our replies to comments 2 and, especially, 5. We feel that we definitely capture tip vortices of the proper circulation and of reasonably correct core size. Although they do not readily stand out in the velocity contour plots in Figures 3 and 4, if we were to plot contours of vorticity or Q-criterion, which we routinely do, they would be very apparent. The third author, M. Churchfield, who performed the computations especially appreciates your concern. His Ph.D. research was in the computation of aircraft trailing vortices, with the motivation of the safety hazards they pose to trailing aircraft. Wind turbines are quite a different situation. With aircraft trailing vortices, the trailing aircraft often encounters them on a flight path aligned with the vortex axis, or may cross them laterally. With wind turbines, only if the aircraft crossed the top or the bottom of the wake, very near the turbine would the flight path be aligned with the vortex axis, but here the encounter would be very brief, and the effect of the entire helix of vortices would make the flow field very different than an aircraft trailing vortex. In fact, the additive effect of the entire helix is that the wind speed above the wake is slightly faster than freestream, and in the wake it is slower than freestream. The rolling motion is canceled by each successive vortex. And it would be highly unlikely, given airspace constraints around wind farms, that an airplane would ever pass so close to the rotor as to feel the tip vortices before they break down. We feel that the much more important flow feature is the wake deficit itself and the turbulence generated by the wake shear layer.

12. Mulinazzi and Zheng (2014) used the near field turbine tip vortex data from wind tunnel measurements and assumed the scaling relationship that governs the decay of vortices to be the same as that of the aircraft wake vortices. The decay of the vortices is driven by the ambient turbulence in their formulation, which may not be as realistic since conceptually the vortex decay should be most influenced by the turbulence field surrounding the tip vortices (or in this case, turbulence level in the wake of the wind turbine itself). The decay of the tip vortices in Mulinazzi and Zheng (2014) may therefore

represent a conservative scenario. The encounter scenario used in Mulinazzi and Zheng (2014) is then for the small aircraft to hit those highly coherent swirling structures that have sharp velocity gradients in the transverse direction relative to the wingspan of the aircraft. This treatment is considered reasonable and conservative (not necessarily a bad thing for commenting on safety), especially if tip vortices have been shown in the wind turbine literature to persist longer than other velocity deficit or turbulent features. However, once again, the computed tip vortex decay may be too conservative in Mulinazzi and Zheng (2014), since the source of wake decay is taken to be the ambient turbulence instead of the turbulence within the velocity deficit region of the turbine wake. If the LES computation by the authors were shown to be capable of capturing the initial generation of the blade tip vortices with the correct range of strength/circulation, and the computational technique is capable of preserving vorticity without artificial numerical dissipation/diffusion of the vortex, and the encounter scenario involves entering the properly computed / surviving tip vortex, and the conclusion is still that roll moment coefficient is considered low relative to a safe baseline, then it would completely satisfy the current reviewer. This reviewer would like to have some assurance that the results are truly due to all of the possibly relevant flow features being properly captured, and the results are not biased by the computed flow field that cannot capture the critical features that may be important for this specific problem. The tip vortex feature may not be as important in traditional wind turbine wake applications such as siting optimization, but it is considered potentially more important than other features in terms of roll upset. If the numerical scheme and associated modeling of the turbine cannot meaningfully duplicate the tip vortex flow field, then this review would suggest that the wording in the conclusion be modified along the phrases used in Wang, White and Barakos (2017). The aforementioned reference essentially pointed out the flow features their model and LIDAR measurements can reveal, and estimated the risk is only based on those features that their flow field data can resolve.

We agree with the reviewer that, for safety, being conservative is prudent, but we also think that we have fulfilled the reviewer's criterion that "If the LES computation by the authors were shown to be capable of capturing the initial generation of the blade tip vortices with the correct range of strength/circulation, and the computational technique is capable of preserving vorticity without artificial numerical dissipation/diffusion of the vortex, and the encounter scenario involves entering the properly computed / surviving tip vortex, and the conclusion is still that roll moment coefficient is considered low relative to a safe baseline, then it would completely satisfy the current reviewer." We have emphasized the references that support the computational capabilities of this model in our replies to comments 2 and 5.

13. This review once again, thank the authors for the effort to advance the knowledge in this field.

We thank the reviewer for their careful and thoughtful reading of our manuscript.

**Response to Anonymous Referee #3**

This manuscript attempts addressing the issue about flight hazards for airplanes flying in the wake of utility-scale wind turbines. The investigation is carried out through LES simulations and the results are interesting, which might deserve to be disseminated. Please find below some comments.

Comments:

1. P1, L1-4: The first part of the abstract should be more focused and detailed on the motivation, procedure and results. This first four lines sound more appropriate for an introduction; indeed, similar information is reported in Sect. 1.

We agree with the reviewer's observation and have added page 1 lines 6-9 to include more information on the procedure:

"Wind-generated lift forces and subsequent rolling moments are calculated for hypothetical aircraft transecting the wake in various orientations. Stably and neutrally stratified cases are explored, with the stable case presenting a possible worst-case scenario due to longer-persisting wakes permitted by lower ambient turbulence."

2. P1, L7: You could add that you deem stable and neutral conditions more critical than convective conditions due to the faster wake recovery. Therefore, you did not performed simulations under convective conditions.

Page 1 lines 7-9 has been amended to read:

"Stably and neutrally stratified cases are explored, with the stable case presenting a possible worst-case scenario due to longer-persisting wakes permitted by lower ambient turbulence."

3. P2, L9 : Why do you only consider roll and not, for instance, pitch moment? Maybe there are safety standards in general aviation that I am not aware of. In that case, please provide related references. I think that a non-symmetric velocity field induced by the wind turbine wake can also affect pitch and yaw of the airplane, which might be a risky situation leading to a premature wing stall. Please comment on this and, eventually, clarify.

The reviewer is correct that pitch and yaw moments could also be hazards for general aviation aircraft encountering wind turbine wakes. We chose to focus time and computational resources on conducting an in-depth analysis of the roll hazard in particular as a direct response to a prior study that found that motion to be especially hazardous (Mulinazzi and Zheng, 2014). Page 2 lines 12-13 and 18-19 now emphasize that we chose to investigate the rolling moment because Wang et al. (2015) cite the rolling moment to be the most concerning to general aviation pilots, and Mulinazzi and Zheng (2014) used the rolling moment as their main hazard identifier:

"General aviation pilots are typically most concerned with the rolling moment (Wang et al., 2015), and we thus focus our study on this hazard…. Previous work has argued that turbine wakes present, in particular, a serious roll hazard to general aviation aircraft. Mulinazzi and Zheng (2014) used a helical vortex model to represent a wind turbine wake from which aircraft roll hazards were calculated."

Additionally, we now emphasize that pitch and yaw hazards may be good avenues for future work in page 12 lines 33-34: "Our conclusions are drawn from rolling moment calculations, though additional calculations of yawing and pitching moments could also be beneficial."

4. P2, L11-12: "The rolling moment is the aerodynamic force applied", a moment cannot be a force. Maybe rephrase it saying that the roll moment is the result of the lift distribution over the wing span.

Thank you – this is an important distinction and page 2 lines 13-15 have been edited to now read: "The rolling moment is the tendency for an aerodynamic force applied at a distance from an aircraft's center of mass to cause the aircraft to undergo angular acceleration about its roll axis, considered a torque in this study". A reference to a flight dynamics textbook explaining our use and form of the rolling moment has also been included for interested readers (Etkin and Reid, 1996).

5. Eq. 5 is inconsistent. \beta should be the wing span, not the aspect ratio. I hope this being only a typo and not jeopardizing your analysis. Please cross-check your data analysis as well.

We thank you for catching this issue and have corrected it. Page 8 lines 1-2 should have said "b is the wingspan of the aircraft, which we set to be 10 m", and we have edited it accordingly. The calculations in the analysis were correct and remain unchanged, but now the misnomer of a variable is fixed.

6. P8, L5-7: "a 10-by-10 array of aircraft", this description is not clear. I think It will be better to talk about aircraft paths rather than aircraft arrays. Please try to rephrase it.

Page 8 lines 16-20 have been re-written for clarification to now read "The down-wake group, 100 aircraft flight paths separated by 15 grid points (~18.5 m) in y and z, is centered over the wake. We march these aircraft downwind from the turbine through the x-extent of the domain. The cross-wake group is comprised of 44 rows and 10 columns of aircraft flight paths separated by 15 grid points (~18.5 m) in x and z initially positioned at y = -2D, the furthest west point of the domain."